# Interpretation and Transformation of Intrinsic Camera Parameters Used in Photogrammetry and Computer Vision

**DOI:** 10.3390/s22249602

**Published:** 2022-12-07

**Authors:** Kuan-Ying Lin, Yi-Hsing Tseng, Kai-Wei Chiang

**Affiliations:** Department of Geomatics, National Cheng Kung University, No. 1, Daxue Road, East District, Tainan City 701, Taiwan

**Keywords:** intrinsic camera parameters, camera mathematical model, camera calibration, photogrammetry, computer vision

## Abstract

The precision modelling of intrinsic camera geometry is a common issue in the fields of photogrammetry (PH) and computer vision (CV). However, in both fields, intrinsic camera geometry has been modelled differently, which has led researchers to adopt different definitions of intrinsic camera parameters (ICPs), including focal length, principal point, radial distortion, decentring distortion, affinity and shear. These ICPs are indispensable for vision-based measurements. These differences can confuse researchers from one field when using ICPs obtained from a camera calibration software package developed in another field. This paper clarifies the ICP definitions used in each field and proposes an ICP transformation algorithm. The originality of this study lies in its use of least-squares adjustment, applying the image points involving ICPs defined in PH and CV image frames to convert a complete set of ICPs. This ICP transformation method is more rigorous than the simplified formulas used in conventional methods. Selecting suitable image points can increase the accuracy of the generated adjustment model. In addition, the proposed ICP transformation method enables users to apply mixed software in the fields of PH and CV. To validate the transformation algorithm, two cameras with different view angles were calibrated using typical camera calibration software packages applied in each field to obtain ICPs. Experimental results demonstrate that our proposed transformation algorithm can be used to convert ICPs derived from different software packages. Both the PH-to-CV and CV-to-PH transformation processes were executed using complete mathematical camera models. We also compared the rectified images and distortion plots generated using different ICPs. Furthermore, by comparing our method with the state of art method, we confirm the performance improvement of ICP conversions between PH and CV models.

## 1. Introduction

The precision modelling of the intrinsic geometry of a camera is essential to the various applications of the vision-based measurements acquired with the visual sensors. Digital cameras are the most widespread visual sensors applied to vision-based measurements due to their generality and affordability. The camera mathematical model is developed based on the intrinsic camera parameters (ICPs). ICPs obtained from the process of camera calibration are a key factor in accurate vision-based measurements in the field of instrumentation and measurement [1]. This is especially true in the fields of photogrammetry (PH) and computer vision (CV). Although both fields consider the intrinsic camera geometry as a perspective projection (pinhole camera model) with the lens distortion model, the adopted ICPs of the associated mathematical models in these two fields differ. Consequently, researchers from one field are frequently confused when using camera parameters obtained from a camera calibration software package developed in the other field. To ameliorate this problem, this paper clarifies the ICP definitions used in each field and proposes an ICP transformation algorithm for greater interoperability between both fields.

In the field of PH, the calibration of metric cameras has been studied since the 1960s [2]. The parameters derived to describe the intrinsic camera geometry are known as interior orientation parameters and are applied to correct image measurements to satisfy the geometry of a perspective projection. During the era of film cameras, the major items of calibration included lens aberrations, film deformation, the focal length and the principal point [3]. In film cameras, the principal point must be determined using fiducial marks fixed inside the camera; such cameras are called metric cameras. In digital cameras, the visual sensor is usually fixed inside the camera, so the principal point can be determined without the need for fiducial marks. Furthermore, the deformation of visual sensors is negligible. Therefore, the focal length, the principal point and lens aberration of digital cameras have become the standard parameters in modelling [4]. Lens aberration in Brown’s [3] model incorporates the radial distortion and decentring distortion. Additional parameters, such as affine distortion [5] and chromatic aberration [6], can also be adopted. Such a calibration model can also be extended to other kinds of cameras, such as a hemispheric camera [7], RGB-D camera [8], thermal camera [9] and line-scan camera [10].

In the field of PH, lens aberrations are modelled as correction equations in which lens distortion terms are a function of distorted image coordinates [3]. The advantage of this method is that image measurements can be directly corrected against lens aberrations using the calibrated lens distortion parameters. This allows corrected image points to be applied in the calculation of bundle adjustment. Through bundle adjustment, 3D coordinates of object points corresponding to the measurements of image tie points can be computed with reasonable precision. In the past, this approach was beneficial to aerial photogrammetry for mapping and surveying [11].

Camera mathematical models have also been developed and utilized in the field of CV. These CV models originated from the field of PH [12,13]. Similarly, ICPs related to lens distortion have been widely adopted and have become the classic approach [14]. The basic mathematical model in CV for a camera is based on a pinhole camera model compensating for lens distortion [13]. Although many variations have been proposed in the literature, almost all of them have been based on this model. This compensation model can be considered as the inverse of the correction model applied in the field of PH. It has been adopted and become the standard in the field of CV without undergoing any major modifications since its introduction [15]. By applying this compensation model, image lens distortion can be efficiently rectified. For the calculation of image rectification, the plane where the coordinates of image points are measured is named the frame buffer, and the plane where normalized image points are located is named the CCD plane [16]. The CCD plane is defined instead as the normalized image plane in the textbook [17]. At present, normalized image points are often used in computation; for example, they may be used to find the essential matrix [18]. To improve the convenience of generating rectified images, lens distortion terms are chosen to be a function of undistorted image coordinates. The advantage of this mathematical model is that the rectified image can be resampled from the original image efficiently without additional iterative calculations.

Wang et al. [19] performed a comparison of the coordinate systems, parameter definitions and the camera models commonly applied in the fields of PH and CV. Based on this comparison, they provide a linear transformation of the camera model defined in PH to that defined in CV. This linear transformation is only applicable to radial distortion parameters lower than the third order; therefore, it would not be suitable for highly distorted camera lenses. Drap and Lefèvre [20] proposed an exact formula for inverse radial lens distortions. However, the inversed radial lens distortion parameters defined in this study cannot be applied directly to common CV software, due to the different definition of image coordinates. Furthermore, this transformation does not include the decentring distortion parameters. Hastedt et al. [21,22] presented a table for the transformation of ICPs commonly used in the fields of PH and CV without laying out the derivation process for the conversion. For the validation, they merely demonstrated the transformation of ICPs from CV model to PH model; the other way around was not validated.

In summary, previous studies have noticed the differences of camera models defined in both fields, and some simplified transformation methods have been provided. For the convenience of applying software packages developed in both fields interactively, further studies are needed to provide a comprehensive interpretation and a complete transformation of ICPs commonly applied in both fields.

Hence, the differing models used in each field give rise to confusion among researchers. Because mathematical camera models in the fields of PH and CV are defined differently, the ICPs applied in these two fields cannot be transformed directly. Some proposed simplified methods use closed-form formulas to transform only the dominant ICPs. Although such methods are convenient, the resulting transformations are incomplete, and the accuracy of the converted parameters may be insufficient. This paper proposed a method for comprehensive transformation between PH and CV camera models. The proposed method can be used to transform a complete set of ICPs, including focal length, principal point, radial distortion, decentring distortion, affinity, and shear. This method would be important for those who are treating camera calibration as a critical issue.

Therefore, this paper had three objectives: (1) explicate the camera mathematical models used in both fields and list the equations for image correction according to their respective ICPs; (2) develop a general method for transforming ICPs between the PH and CV models; (3) discuss and analyze the transformed ICP values, rectified images and distortion plots. The three main contributions of this paper are as follows. First, mathematical camera models in the fields of PH and CV are both analyzed in detail. The related equations of ICPs are listed explicitly. Second, a ICP transformation algorithm is proposed. The algorithm uses least-squares adjustment to convert the ICPs by using image points involving ICPs defined in PH and CV image frames. This solves the main problems associated with conventional methods based on simplified formulas. The ICPs can be converted completely between fields of PH and CV, so that the number of converted ICPs obtained using the proposed method is identical to the original number. Selecting suitable image points can improve the accuracy of the generated adjustment model. Finally, this paper summarizes the results of testing the proposed transformation algorithm bidirectionally between the PH and CV models to evaluate the algorithm’s performance.

## 2. Camera Mathematical Model

### 2.1. Geometry of Perspective Projection Used in the Field of PH

The geometry of the perspective projection, as defined in the field of PH, is depicted in Figure 1a. The three coordinate systems (frames) involved in the definition are the object frame, image frame and camera frame. The coordinates of the object points are defined in the object frame. The corresponding coordinates of the image points defined in the image frame are the row (r) and column (c). The unit is the pixel. Figure 1a displays the coordinate axes of the camera frame, whose origin is the perspective center. The *z*-axis represents the optical axis of the camera lens, and the principal point is the intersection of the optical axis and image frame. The distance from the perspective center to the principal point is usually named the principal distance (f), which is the focal length (f) for an infinity focused camera lens. The principal point of the camera is usually close to the center of the image. In this paper, the image coordinates of the principal point are denoted ($c_{p},\;r_{p}$), as indicated in Figure 1b.

Based on the preceding definition, the coordinates of an image point in the image frame are $\left(c,\;r\right)$ and the corresponding coordinates in the camera frame are (x, y,−f). According to the frame relationship, the transformation from image frame to camera frame can be presented in the form of homogeneous coordinates. Equation (1) describes the transformation process. Metric units are conventionally adopted for the PH camera frame. Therefore, the formula incorporates the image pixel size (*ds*).
(1)$$\left[\begin{array}{c} x\\ y\\ -f \end{array}\right]=\left[\begin{array}{ccc} ds & 0 & -ds\cdot c_{p}\\ 0 & -ds & ds\cdot r_{p}\\ 0 & 0 & -f \end{array}\right]\cdot \left[\begin{array}{c} c\\ r\\ 1 \end{array}\right]$$

There is another definition of the image frame in the field of PH. Two definitions of image frame are depicted in Figure 2. The origin is set on the image center, which is denoted ($c_{c},\;r_{c}$), as indicated in Figure 2b. Metric units are adopted in this definition. The coordinates of an image point in this image frame are denoted $\left(x_{i},\;y_{i}\right)$, and the coordinates of principal point are denoted ($x_{p},\;y_{p}$). The conversion of principal point in two image frames is listed in Equation (2). The corresponding coordinates in the camera frame are still (x, y,−f) that are calculated from Equation (3).
(2)$$\left[\begin{array}{c} c_{p}\\ r_{p} \end{array}\right]=\left[\begin{array}{c} c_{c}+\frac{x_{p}}{ds}\\ r_{c}-\frac{y_{p}}{ds} \end{array}\right]$$
(3)$$\left[\begin{array}{c} x\\ y \end{array}\right]=\left[\begin{array}{c} x_{i}-x_{p}\\ y_{i}-y_{p} \end{array}\right]$$

For digital cameras, it would be more convenient to use the first definition than the second definition. In this paper, we adopt the first definition of image frame in the field of PH. Therefore, even if the image frame in the calibration software used is defined in metric unit, the coordinates of image point can be converted into the image frame as defined in pixel unit.

The preceding transformation is incomplete if lens distortion is uncorrected. In the camera frame, the coordinates of an image point with lens distortion should be denoted ($x_{d},\;y_{d},\;-f$) instead of (x, y,−f). Camera lens distortion has been commonly treated as a combination of radial and decentring (tangential) distortion, which are modelled as functions of distorted coordinates in the field of PH [3]. Therefore, the undistorted coordinates ($x_{u},\;y_{u},\;-f$) can be obtained by adding the radial and decentring distortion terms to the distorted coordinates. Equation (4) displays the lens distortion model, in which ($\Delta x_{rad},\;\Delta y_{rad}$) denotes radial distortion and ($\Delta x_{dec},\;\Delta y_{dec}$) denotes decentring distortion.
(4)$$\left[\begin{array}{c} x_{u}\\ y_{u} \end{array}\right]=\left[\begin{array}{c} x_{d}+\Delta x_{rad}\left(x_{d},y_{d}\right)+\Delta x_{dec}\left(x_{d},y_{d}\right)\\ y_{d}+\Delta y_{rad}\left(x_{d},y_{d}\right)+\Delta y_{dec}\left(x_{d},y_{d}\right) \end{array}\right]$$

In general, a camera lens has a much larger amount of radial distortion than decentring distortion. Radial distortion can be modelled using a polynomial function with multiple parameters, such as $k_{1}$, $k_{2}$, $k_{3}$, etc. [3]. Equation (5) lists the formulas of radial distortion. Three parameters are sufficient in most cases, with one exception being the case of fisheye lens cameras. Brown [3] also proposed a mathematical function for decentring distortion with two parameters, $p_{1}$ and $p_{2}$. Equation (6) lists the formula of decentring distortion.
(5)$$\left[\begin{array}{c} \Delta x_{rad}\left(x_{d},y_{d}\right)\\ \Delta y_{rad}\left(x_{d},y_{d}\right) \end{array}\right]=\left[\begin{array}{c} x_{d}\left(k_{1}r^{2}+k_{2}r^{4}+k_{3}r^{6}\right)\\ y_{d}\left(k_{1}r^{2}+k_{2}r^{4}+k_{3}r^{6}\right) \end{array}\right]$$
(6)$$\left[\begin{array}{c} \Delta x_{dec}\left(x_{d},y_{d}\right)\\ \Delta y_{dec}\left(x_{d},y_{d}\right) \end{array}\right]=\left[\begin{array}{c} p_{1}\left(r^{2}+2x_{d}{}^{2}\right)+2p_{2}x_{d}y_{d}\\ 2p_{1}x_{d}y_{d}+p_{2}\left(r^{2}+2y_{d}{}^{2}\right) \end{array}\right]$$
where $r=\sqrt{x_{d}^{2}+y_{d}^{2}}$.

There are the additional parameters of affinity and shear to take into account. However, such terms are rarely significant for common digital cameras [23]. The affinity and shear represent image-invariant affine distortion [4]. Equation (7) lists the formulas of affine distortion. The parameter of affinity is denoted $b_{1}$. The parameter of shear is denoted $b_{2}$. Affine distortion can be added with radial and decentring distortion into Equation (4).
(7)$$\left[\begin{array}{c} \Delta x_{aff}\left(x_{d},y_{d}\right)\\ \Delta y_{aff}\left(x_{d},y_{d}\right) \end{array}\right]=\left[\begin{array}{c} b_{1}x_{d}+b_{2}y_{d}\\ 0 \end{array}\right]$$

These additional parameters are mainly relevant to older frame grabber-based cameras, in which timing problems often lead to horizontal pixel-spacing relative errors and image shearing [5]. Therefore, these additional parameters are often ignored for digital cameras due to insignificant effect. In this paper, we also ignore the parameters of affinity and shear in the field of PH, so that $b_{1}$ and $b_{2}$ are set to zero.

### 2.2. Geometry of Perspective Projection Used in the Field of CV

The geometry of perspective projection, as defined in the field of CV, is depicted in Figure 3a. The coordinate systems involved include the object, image and camera frames, but the axes of the camera frame are defined differently from those in the field of PH. Furthermore, pixel units rather than metric units are adopted. This causes the transformation matrix from the image frame to the camera frame to differ from that used in the field of PH. The corresponding coordinates of the image points defined in the image frame are also the row and column. The unit is the pixel. The image coordinates of the principal point are also denoted ($c_{p},\;r_{p}$), as indicated in Figure 3b.

In addition, the image pixel shape is not always modelled as a perfect square in the field of CV. The lengths of the image pixel in the *x* and *y* direction are represented by $ds_{x}$ and $ds_{y}$, respectively. Therefore, the *x* and *y* coordinates of the focal length are termed $f_{x}$ and $f_{y}$. CV researchers often prefer using homogeneous coordinate transformation, in which case the images defined in the camera frame are represented with normalized homogeneous coordinates. In this paper, this is termed the normalized image frame, as indicated in Figure 4.

In the normalized image frame, the coordinate unit is normalized to be the focal length. Accordingly, the homogeneous coordinates of each image point in the camera frame become ($\hat{x},\;\hat{y},1$). Therefore, the transformation from the image frame to the camera frame can be presented in a form of homogeneous coordinates in Equation (8). The parameters of affinity and shear are presented in this equation. The parameter of shear is denoted s. The parameter of affinity is implicit in $f_{x}$ and $f_{y}$ that is presented in Equation (9).
(8)$$\left[\begin{array}{c} \hat{x}\\ \hat{y}\\ 1 \end{array}\right]=\left[\begin{array}{ccc} \frac{1}{f_{x}} & \frac{-s\hat{y}}{f_{x}r} & \frac{-c_{p}}{f_{x}}\\ 0 & \frac{1}{f_{y}} & \frac{-r_{p}}{f_{y}}\\ 0 & 0 & 1 \end{array}\right]\cdot \left[\begin{array}{c} c\\ r\\ 1 \end{array}\right]$$
(9)$$\left\{\begin{array}{c} f_{x}=\frac{f}{ds_{x}}\\ f_{y}=\frac{f}{ds_{y}} \end{array}\right.$$

In the field of CV, the term skew is more often used than shear. For consistency in this paper, we still adopt the term shear as used in the field of PH. Since shear is usually very small, the assumption that s=0, commonly used by other authors [15,24], is quite reasonable [25]. In reality, the shear might not be zero, because when taking an image of an image [18], the *x*- and *y*- axes are not perpendicular. Therefore, the parameter of shear is often ignored for digital cameras due to insignificant effect. In this paper, we also ignore the parameter of shear in the field of CV, so that s is set to zero and Equation (8) is simplified to Equation (10).
(10)$$\left[\begin{array}{c} \hat{x}\\ \hat{y}\\ 1 \end{array}\right]=\left[\begin{array}{ccc} \frac{1}{f_{x}} & 0 & \frac{-c_{p}}{f_{x}}\\ 0 & \frac{1}{f_{y}} & \frac{-r_{p}}{f_{y}}\\ 0 & 0 & 1 \end{array}\right]\cdot \left[\begin{array}{c} c\\ r\\ 1 \end{array}\right]$$

Although researchers in both fields apply Brown’s formulas and use similar parameters, those parameters are defined differently. Therefore, software packages yield very different lens distortion parameter values for camera calibration, depending on whether that package was developed for PH or CV. In the field of CV, lens distortion is modelled in the undistorted image points rather than in the distorted points. The transformed coordinates generated from Equation (10) should contain distortion and can be denoted as (${\hat{x}}_{d},\;{\hat{y}}_{d},\;1$). The undistorted coordinates (${\hat{x}}_{u},\;{\hat{y}}_{u},\;1$) are obtained by subtracting the radial and decentring distortion terms from the distorted coordinates. Therefore, lens distortion is modelled as Equation (11) in the field of CV.
(11)$$\left[\begin{array}{c} {\hat{x}}_{u}\\ {\hat{y}}_{u} \end{array}\right]=\left[\begin{array}{c} {\hat{x}}_{d}-\Delta {\hat{x}}_{rad}\left({\hat{x}}_{u},{\hat{y}}_{u}\right)-\Delta {\hat{x}}_{dec}\left({\hat{x}}_{u},{\hat{y}}_{u}\right)\\ {\hat{y}}_{d}-\Delta {\hat{y}}_{rad}\left({\hat{x}}_{u},{\hat{y}}_{u}\right)-\Delta {\hat{y}}_{dec}\left({\hat{x}}_{u},{\hat{y}}_{u}\right) \end{array}\right]$$

Equations (12) and (13) are the mathematical functions applied for radial distortion and decentring distortion, respectively, in the field of CV. The parameters for radial distortion, ${\hat{k}}_{1}$, ${\hat{k}}_{2}$ and ${\hat{k}}_{3}$, and for decentring distortion, ${\hat{p}}_{1}$ and ${\hat{p}}_{2}$, appear similarly to those applied in the field of PH, but they are defined fairly differently. First, they are defined in the normalized image frame and are a function of the undistorted coordinates. This makes the parameter transformation between these two definitions non-transparent. Second, ${\hat{p}}_{1}$ and ${\hat{p}}_{2}$ are also defined differently.
(12)$$\left[\begin{array}{c} \Delta {\hat{x}}_{rad}\left({\hat{x}}_{u},{\hat{y}}_{u}\right)\\ \Delta {\hat{y}}_{rad}\left({\hat{x}}_{u},{\hat{y}}_{u}\right) \end{array}\right]=\left[\begin{array}{c} {\hat{x}}_{u}\left({\hat{k}}_{1}{\hat{r}}^{2}+{\hat{k}}_{2}{\hat{r}}^{4}+{\hat{k}}_{3}{\hat{r}}^{6}\right)\\ {\hat{y}}_{u}\left({\hat{k}}_{1}{\hat{r}}^{2}+{\hat{k}}_{2}{\hat{r}}^{4}+{\hat{k}}_{3}{\hat{r}}^{6}\right) \end{array}\right]$$
(13)$$\left[\begin{array}{c} \Delta {\hat{x}}_{dec}\left({\hat{x}}_{u},{\hat{y}}_{u}\right)\\ \Delta {\hat{y}}_{dec}\left({\hat{x}}_{u},{\hat{y}}_{u}\right) \end{array}\right]=\left[\begin{array}{c} 2{\hat{p}}_{1}{\hat{x}}_{u}{\hat{y}}_{u}+{\hat{p}}_{2}\left({\hat{r}}^{2}+2{\hat{x}}_{u}^{2}\right)\\ {\hat{p}}_{1}\left({\hat{r}}^{2}+2{\hat{y}}_{u}^{2}\right)+2{\hat{p}}_{2}{\hat{x}}_{u}{\hat{y}}_{u} \end{array}\right]$$
where $\hat{r}=\sqrt{{\hat{x}}_{u}^{2}+{\hat{y}}_{u}^{2}}$.

There are a number of parameters, variables, and constants in the mathematical expressions. Table 1 lists their symbols and corresponding meanings, as defined in PH and CV standard.

## 3. Methodology

As mentioned, the transformation of ICPs is conventionally nonlinear. If transformation from one camera coordinate system to another is required, the conventional method is linearisation for related equations with iterative computation. The process would be complicated and computationally expensive. The initial value for linearisation must also be determined. However, image point coordinates can be transformed linearly. The coordinates of image points involving ICPs are the same defined in PH and CV image frames. Based on this principle, we propose a linear transformation theory for converting ICPs between the PH and CV camera mathematical models. The least-squares adjustment is used in our transformation algorithm. The corresponding weight of each observation is set to the same. The main challenges that must be addressed in ICP transformation are as follows: (1) selecting suitable image points in the PH or CV camera frame, accounting for both the number and distribution of image points; (2) transforming image points between the PH camera frame and the CV camera frame; and (3) using least-squares adjustment to transform a complete set of ICPs. The means by which the proposed method addresses these challenges are detailed in the following sections.

### 3.1. Transformation of ICPs from PH to CV Standard

The five-step workflow for transforming PH ICPs to CV ICPs is depicted in Figure 5. Each step is described in Figure 5a, and the entire framework is visualized in Figure 5b. The first step involves the selection of an adequate number of distorted image points as observation points. According to the image points selected, corresponding undistorted image points can be calculated based on ICPs by using Equation (4) to Equation (6). The coordinate system of distorted image points and undistorted image points are defined in the PH camera frame. Thus, the second step involves the calculation of the corresponding undistorted image points. The procedure was described as in the section ‘Geometry of Perspective Projection Used in the Field of PH’. The proposed ICP transformation method requires at least three observed image points to generate a solution. However, the solution generated when the minimum number of points is used may not be reliable. A reliable solution requires not only an adequate number of observation points, but also an even distribution of observation points. Selecting observation points at a particular intervals across the whole image may be a suitable strategy for achieving an even distribution. This can ensure that the most reliable solution can be obtained even when the number of observation points is limited.

The third step involves the transformation of image points from the PH camera frame to the CV camera frame. Both the undistorted and distorted image points must be converted in this step. The two parts of the step are detailed as follows. The first part is a coordinate transformation from the camera frame to the image frame. These two coordinate systems are defined in the field of PH. The equation obtained from Equation (1) is listed in Equation (14).
(14)$$\left[\begin{array}{c} c\\ r\\ 1 \end{array}\right]=\left[\begin{array}{ccc} \frac{1}{ds} & 0 & \frac{c_{p}}{-f}\\ 0 & \frac{-1}{ds} & \frac{r_{p}}{-f}\\ 0 & 0 & \frac{1}{-f} \end{array}\right]\cdot \left[\begin{array}{c} x\\ y\\ -f \end{array}\right]$$

The second part is a coordinate transformation from the camera frame to the image frame. These two coordinate systems are defined in the field of CV. The equation obtained from Equation (10) is listed in Equation (15).
(15)$$\left[\begin{array}{c} c\\ r\\ 1 \end{array}\right]=\left[\begin{array}{ccc} f_{x} & 0 & c_{p}\\ 0 & f_{y} & r_{p}\\ 0 & 0 & 1 \end{array}\right]\cdot \left[\begin{array}{c} \hat{x}\\ \hat{y}\\ 1 \end{array}\right]$$

The coordinates of the image point in the image frame are the same in the PH and CV models; Equations (16) and (17) are derived from Equations (14) and (15). Based on these two equations, the coordinates of the undistorted and distorted images points in the CV camera frame can be obtained. Therefore, coordinate transformation from the PH camera frame to the CV camera frame can be achieved. The conversion of the focal length from the field of PH to the field of CV is defined in Equation (18).
(16)$$\hat{x}=\frac{x}{ds\cdot f_{x}}$$
(17)$$\hat{y}=\frac{-y}{ds\cdot f_{y}}$$
(18)$$f_{x}=f_{y}=\frac{f}{ds}$$

The fourth step involves the listing of the observation equations that are reorganized based on the correction equations in the field of CV. They are listed as Equations (11)–(13). The unknown parameters are ICPs in the field of CV. Equation (19) presents the entire least-squares form for solving CV ICPs. The undistorted image points are listed from (${\hat{x}}_{u1},\;{\hat{y}}_{u1}$) to (${\hat{x}}_{un},\;{\hat{y}}_{un}$) sequentially. Distorted image points are listed from (${\hat{x}}_{d1},\;{\hat{y}}_{d1}$) to (${\hat{x}}_{dn},\;{\hat{y}}_{dn}$) sequentially. All coordinates of the image points are known. Every value in the design matrix can be computed directly as coefficients. Consequently, this adjustment model is linear.
(19)$$\left[\begin{array}{c} \begin{array}{c} {\hat{x}}_{d1}-{\hat{x}}_{u1}\\ {\hat{y}}_{d1}-{\hat{y}}_{u1} \end{array}\\ \begin{array}{c} \vdots \\ \begin{array}{c} {\hat{x}}_{dn}-{\hat{x}}_{un}\\ {\hat{y}}_{dn}-{\hat{y}}_{un} \end{array} \end{array} \end{array}\right]+V=\left[\begin{array}{ccc} {\hat{x}}_{u1}{\hat{r}}^{2} & {\hat{x}}_{u1}{\hat{r}}^{4} & {\hat{x}}_{u1}{\hat{r}}^{6}\\ {\hat{y}}_{u1}{\hat{r}}^{2} & {\hat{y}}_{u1}{\hat{r}}^{4} & {\hat{y}}_{u1}{\hat{r}}^{6}\\ {\hat{x}}_{un}{\hat{r}}^{2} & {\hat{x}}_{un}{\hat{r}}^{4} & {\hat{x}}_{un}{\hat{r}}^{6}\\ {\hat{y}}_{un}{\hat{r}}^{2} & {\hat{y}}_{un}{\hat{r}}^{4} & {\hat{y}}_{un}{\hat{r}}^{6} \end{array}\vdots \begin{array}{cc} \;\;\;2{\hat{x}}_{u1}{\hat{y}}_{u1} & {\hat{r}}^{2}+2{\hat{x}}_{u1}\\ \;\;{\hat{r}}^{2}+2{\hat{y}}_{u1} & 2{\hat{x}}_{u1}{\hat{y}}_{u1}\\ \;\;2{\hat{x}}_{un}{\hat{y}}_{un} & {\hat{r}}^{2}+2{\hat{x}}_{un}\\ \;\;{\hat{r}}^{2}+2{\hat{y}}_{un} & 2{\hat{x}}_{un}{\hat{y}}_{un} \end{array}\right]\left[\begin{array}{c} {\hat{k}}_{1}\\ \begin{array}{c} {\hat{k}}_{2}\\ {\hat{k}}_{3}\\ \begin{array}{c} {\hat{p}}_{1}\\ {\hat{p}}_{2} \end{array} \end{array} \end{array}\right]$$
where V denotes the residual vector.

The entire least-squares adjustment can be expressed as L+V=A·X, where L represents the matrix of observations, A represents the design matrix, and X represents the matrix of unknown parameters. Once the observation equations have been listed sequentially, the related values in the matrix of observations and design matrix can be calculated accordingly. X can be directly solved as ${\left(A^{T}\cdot A\right)}^{-1}\cdot A^{T}\cdot L$.

### 3.2. Transformation of ICPs from CV to PH Standard

The five-step workflow is depicted in Figure 6. Each step is described in Figure 6a, and the entire framework is visualized in Figure 6b.

The process of transforming CV ICPs to PH ICPs is highly similar to the aforementioned process of transforming PH ICPs to CV ICPs. The procedure also comprises five steps. The major difference is that undistorted rather than distorted image points must be selected in the first step. Subsequently, in the second step, the corresponding distorted image points can be calculated based on ICPs. The equations are listed in Equations (11)–(13). The coordinate system of distorted image points and undistorted image points is defined in the CV camera frame.

The third step involves the transformation of image points from the CV camera frame to the PH camera frame. Both undistorted and distorted image points must be converted in this step. Equations (20) and (21), which are derived from Equations (16) and (17), detail the process. The conversion of the focal length from the CV to PH standard is defined in Equation (22). For the parameter of another focal length in CV standard, $f_{x}$, its effect can be transformed into the parameter of shear, $b_{1}$ in PH standard. Therefore, after the transformation of CV ICPs to PH ICPs, the number of parameters is still the same.
(20)$$x=\hat{x}\cdot ds\cdot f_{x}$$
(21)$$y=-\hat{y}\cdot ds\cdot f_{y}$$
(22)$$f=f_{y}\cdot ds_{y}$$

The fourth and final step involves listing the observation equations that are reorganized on the basis of Equations (4)–(7). The unknown parameters are ICPs defined according to the PH standard. Equation (23) displays the entire least-squares form for solving PH ICPs. Distorted image points are listed sequentially from ($x_{d1},\;y_{d1}$) to ($x_{dn},\;y_{dn}$). Undistorted image points are listed sequentially from ($x_{u1},\;y_{u1}$) to ($x_{un},\;y_{un}$). All coordinates of the image points are known. Every value in the design matrix can be computed directly as a coefficient. Consequently, this adjustment model is linear as well. If the affinity factor in Equation (7) is ignored, this adjustment model can be simplified by excluding the $b_{1}$ parameter in Equation (23). Least-squares adjustment can be expressed as the equation L+V=A·X. Using this equation, the converted ICPs can be solved.(23)$$\left[\begin{array}{c} \begin{array}{c} x_{u1}-x_{d1}\\ y_{u1}-y_{d1} \end{array}\\ \begin{array}{c} \vdots \\ \begin{array}{c} x_{un}-x_{dn}\\ y_{un}-y_{dn} \end{array} \end{array} \end{array}\right]+V=\left[\begin{array}{cccccc} x_{d1}r^{2} & x_{d1}r^{4} & x_{d1}r^{6} & r^{2}+2x_{d1}{}^{2} & 2x_{d1}y_{d1} & x_{d1}\\ y_{d1}r^{2} & y_{d1}r^{4} & y_{d1}r^{6} & 2x_{d1}y_{d1} & r^{2}+2y_{d1}{}^{2} & 0\\ \vdots & \vdots & \vdots & \vdots & \vdots & \vdots \\ x_{dn}r^{2} & x_{dn}r^{4} & x_{dn}r^{6} & r^{2}+2x_{dn}{}^{2} & 2x_{dn}y_{dn} & x_{dn}\\ y_{dn}r^{2} & y_{dn}r^{4} & y_{dn}r^{6} & 2x_{dn}y_{dn} & r^{2}+2y_{dn}{}^{2} & 0 \end{array}\right]\left[\begin{array}{c} k_{1}\\ \begin{array}{c} k_{2}\\ k_{3}\\ \begin{array}{c} p_{1}\\ \begin{array}{c} p_{2}\\ b_{1} \end{array} \end{array} \end{array} \end{array}\right]$$
where V denotes the residual vector.

## 4. Experimental Results

The test cameras, a Sony A6000 (Sony Group Corporation, Tokyo, Japan) and GoPro Hero 4 (GoPro, Inc., San Mateo, CA, USA), are characterized by different degrees of lens distortion. The specifications of these two cameras are listed in Table 2. The cameras were calibrated using typical camera calibration software, respectively applied in the fields of PH and CV to demonstrate the differences between the ICPs, which are produced during the calibration processes in the different fields. Second, to verify the proposed method, both the PH-to-CV and CV-to-PH transformations were executed. In addition, different image point selection strategies were applied to test their effects on the performance of the proposed transformation algorithm.

### 4.1. Camera Calibration Method

The camera calibration methods used in the fields of PH and CV differ. The technique in the field of PH is more rigorous, and includes a greater number of camera parameters [26]; whereas, the technique in the field of CV emphasises automation, efficiency and versatility [27]. Remondino and Fraser [23] performed a detailed review of the calibration approaches used in the fields of PH and CV. In this paper, only one method was selected for each camera mathematical model, to reduce complexity. A rotatable table with coded targets [28] and Australis photogrammetric software (version 8.0, Photometrix, Melbourne, Australia) [29] were used for PH camera calibration. A checkerboard [24] and the Camera Calibrator app in MATLAB (version R2021b, The MathWorks, Inc., Natick, MA, USA) [30] were used for CV camera calibration. Figure 7 displays the representative camera calibration methods from the fields of PH and CV used in this paper.

Although the definition of image frame in Australis photogrammetric software is not the same as the one adopted in the paper. Based on Equations (2) and (3), the related coordinates of image points and principal point have been converted to the image frame as defined in pixel unit. Therefore, the proposed method for transforming ICPs can still be implemented.

Table 3 lists the calibration results of the Sony camera. Table 4 lists the calibration results of the GoPro camera. The focal length in the PH calibration method was measured in millimeters, whereas in the CV calibration method, they were measured in units. The units of the lens distortion parameters also differed between the two camera mathematical models. As shown in Table 3 and Table 4, the PH and CV models had millimeter-derived units and a generic ‘unit’, respectively. As mentioned in the previous section, this unit is related to the focal length. In addition, according to Equation (7), $b_{1}$ and $b_{2}$ are only coefficients, so there is not unit.

Consequently, the two types of ICPs and their standards of precision are not the same. Values of ICPs in the same mathematical camera model can be compared, but the precision of ICPs from different models cannot be compared. Nevertheless, the overall precision in both the PH and CV calibration results was less than 1.5 pixels, according to the calibration reports in our experiments. For the Sony camera, the overall precision achieved when the PH and CV calibration methods were applied was 0.27 and 0.71 pixels, respectively. For the GoPro camera, the overall precision achieved when the PH and CV calibration methods were applied was 0.30 and 1.48 pixels, respectively.

### 4.2. ICP Transformation Results: From the PH to CV Standard

The proposed algorithm uses several image point selection strategies to account for the effects of the number and distribution of selected observation points. The first strategy involves three selected observation points with an intensive distribution (Case 1). The second involves three selected observation points with an extensive distribution (Case 2). The third involves 12 selected observation points evenly distributed across the whole image (Case 3). The last involves 4800 selected observation points placed at regular intervals across the whole image, ensuring an even distribution (Case 4). The selected observation points of Cases 1 to 4, when the Sony camera was used, are displayed in Figure 8. The experimental design adopted when the GoPro camera was used was identical. We implemented these strategies and compared four sets of results with the original calibration results obtained using the CV method.

#### 4.2.1. Sony A6000 Results

Table 5 displays the Case 1 to 4 results obtained when the Sony camera was used. The focal length and principal point values obtained from the calibration and conversion methods differed by a maximum of 12 pixels. Overall, each ICP value was similar between the different methods, indicating that the proposed transformation algorithm is feasible.

Notably, $k_{1}$ was negative for the CV method results and positive for the PH method results. Normally, $k_{1}$ is positive for PH method results for most cameras due to barrel distortion. These two camera mathematical models are inversely related. Therefore, the range of $k_{1}$ in the converted results is the inverse of the original range. However, the other distortion parameters, including $k_{2},$ $k_{3}$, $p_{1}$ and $p_{2}$, may not accord with the aforementioned condition because they are much less influential than $k_{1}$.

The numerical results in Table 5 do not clearly indicate which set of ICPs is optimal. Therefore, we generated rectified images using ICPs obtained from each method. The effects of rectification (with relevant image resolutions listed) are illustrated in Figure 9. The differences were most visible in the four corners of each image. The distortion correction in Case 1 was more obvious than that in the other cases; however, the five rectified images look similar to the original images because of the small amount of lens distortion. Moreover, the differences in the resolution of the images, rectified using different methods, were minimal. The difference in resolution between the original image and each rectified images was less than 60 pixels.

The radial distortion and decentring distortion plots generated with the CV model are displayed in Figure 10. For each radial distortion plot, the red, blue and pink solid lines indicate $k_{1}$, $k_{2}$ and $k_{3}$, respectively. The black dotted line indicates the overall radial distortion curve. For each decentring distortion plot, the red and blue solid lines indicate $p_{1}$ and $p_{2}$, respectively. The black dotted line indicates the overall decentring distortion curve. The values in the *x-* and *y*-axes in each figure are in pixels. In both plots, the boundary of *x*-axis is 3500 pixels. The curves obtained using the CV method can be used as a reference to evaluate the fit of the curves in Cases 1 to 4.

For the radial distortion plot, the range of the *y*-axis is from −100 to 100 pixels. Although only three observation points were selected in Cases 1 and 2, the curves in Case 1 were steeper, indicating that the distortion of the image points closer to the edge is greater. The curves in Case 2 were consistent with the curves obtained using the CV method. Therefore, using three observation points with an extensive distribution is a more feasible transformation strategy. The curves in Cases 3 and 4 were nearly identical, indicating that using 12 observation points with an even distribution can achieve transformation results equivalent to those that can be obtained using 4800 observation points. The curves in Cases 3 and Case 4 and the curved obtained using the CV method were similar. For all methods, the $k_{1}$ curve decreased. As mentioned, the PH and CV models are inversely related; this is evident in the radial distortion plot. For the decentring distortion plot, the range of the *y*-axis is from −0.001 to 0.001 pixels. Although the difference between each curve was small, in general, the curve for the CV method had the steepest gradient.

In summary, the number of selected observation points and distribution both affect the transformation of ICPs. After these two factors were accounted for, the curves in Case 4 had the smallest gradients, indicating that the corresponding method achieved the most favorable results. This indicates that ICPs converted based on image points selected at a set interval across the whole image are more suitable for the Sony camera. These ICPs also yielded better results than the original ICPs calibrated by the checkerboard.

#### 4.2.2. GoPro Hero 4 Results

The results of Cases 1 to 4 for the GoPro camera are listed in Table 6. Focal length values obtained from the calibration and conversion methods differed greatly. The difference was approximately 67 pixels. By contrast, the principal point values differed slightly. Furthermore, the radial distortion of the GoPro camera belongs to barrel distortion, so that $k_{1}$ was also negative in the CV method. The results of the four cases were not completely consistent. Nevertheless, the corresponding ICP values acquired using the different methods were still similar, demonstrating the feasibility of our transformation algorithm.

Again, we generated rectified images using ICPs to evaluate which set of ICPs was superior. The rectified images and their corresponding resolutions are presented in Figure 11.

Because the lens distortion of the GoPro camera was much larger, the effects of rectification were much more noticeable. The rectified image in Case 1 was considerably deformed, but the rectified image in Case 2 was similar to the image rectified using the CV method, indicating that if only three observation points are selected, the distribution of the points is crucial. However, in the image in Case 2 and the image obtained using the CV method, distortion was still visible in the corners; for example, the images depicted three traffic cones instead of two, and a twisted building. Regarding the rectified image Case 3, although some slight distortion remained in the part of the image depicting the road and tree, the rectified image was of higher quality than the original image and similar to the rectified image in Case 4. The image in Case 4 still exhibited the least distortion, indicating that the number of observation points is also a key factor in transformation. In addition, the resolution of the images rectified by these methods differed from that of their corresponding original images by at least 1300 pixels.

The radial distortion and decentring distortion plots defined in the field of CV are depicted in Figure 12. For each radial distortion plot, the red, blue and pink solid lines indicate $k_{1}$, $k_{2}$ and $k_{3}$, respectively. The black dotted line indicates the overall radial distortion curve. For each decentring distortion plot, the red and blue solid lines indicate $p_{1}$ and $p_{2}$, respectively. The black dotted line indicates the overall decentring distortion curve. The values on the *x-* and *y*-axes in each figure are in pixels. The boundary of the *x*-axis is 2500 pixels.

For the radial distortion plot, the range of the *y*-axis is −500 to 500 pixels. Because the distortion of the GoPro camera was larger, the overall curve decreased earlier than the curve for the Sony camera. The curves in Case 1 were steeper than those in Case 2 because of the larger $k_{1}$ and $k_{2}$ values. The curves in Case 2 look like the curves obtained using the CV method. Therefore, an extensive distribution of observation points is more effective for transformation. The curves in Case 4 look like those in Case 3. This indicates that using 12 evenly distributed observation points can achieve transformation results similar to those that can be obtained using 4800 observation points. Overall, the $k_{1}$, $k_{2}$, and $k_{3}$ curves in Case 4 were smoother than those in the other cases and those obtained using the CV method. As indicated in Figure 11, the ICPs in Case 4 were sufficient for the accurate rectification of the images. For the decentring distortion plot, the range of the *y*-axis is from −0.001 to 0.001 pixels. The overall curve in Case 4 was still significantly smoother than the others.

In summary, the number and distribution of the observation points must both be considered in the transformation of ICPs. The results obtained in Case 4 were superior to those obtained in the other cases in terms of radial distortion and decentring distortion. This indicates that ICPs converted based on image points selected at a set interval across the whole image are more suitable for the GoPro camera. These ICPs were also better than the original ICPs calibrated by the checkerboard.

#### 4.2.3. Comparison with the State of Art

To validate the performance, the proposed method is compared with the state of art method presented by Hastedt et al. [21,22]. The Case 4 results of the proposed method are adopted for the comparison. Table 7 and Table 8 list the transformed ICPs resulting from both methods, with respect to Sony and GoPro cameras. The rectified images generated by applying both methods are also shown in Figure 13 and Figure 14, respectively.

Table 7 and Table 8 show that the values of ${\hat{k}}_{1}$ resulting from two methods are apparently different. Compared with the CV calibration results for validation, the values of ${\hat{k}}_{1}$ for both cameras should be negative rather than positive. If we take the difference between the converted ${\hat{k}}_{1}$ value and the validated one, using our method results in differences of 0.0130 and 0.0479 for the Sony and GoPro cameras, respectively. However, the differences are 0.1478 and 0.5534 when using the state of art method. Therefore, the improvement percentages of transformation are 91.2% and 91.3% for the Sony and GoPro cameras, respectively. The values of ${\hat{k}}_{2}$ and ${\hat{k}}_{3}$ are also quite different from the validated values. It is noteworthy that this improvement would affect the image rectification significantly. By checking the rectified images resulting from both methods, shown in Figure 13 and Figure 14, we can see apparent improvement in image rectification when using our method, especially when applied to the GoPro camera. In conclusion, our method outperforms the state of art method when a camera with severe lens distortion is applied, although it needs more computational effort.

### 4.3. ICP Transformation Results: From the CV to PH Standard

Our results indicate that the interval selection of image points is preferred. Therefore, only the interval selection strategy was included for comparison in the part of the experiment described in this section. ICPs of images from the Sony camera and GoPro camera, generated with the CV model, were transformed into PH-type ICPs. Table 9 lists the results for the image from the Sony camera. Table 10 lists the results for the image from the GoPro camera.

In the results of these two tables, the converted ICPs were comparable to the original ICPs. We also conducted the simplified adjustment model in which the affinity factor is ignored. The converted ICPs remain the same, even if $b_{1}$ is not included for the conversion. This demonstrates the feasibility of our proposed algorithm. For the Sony camera, the focal length, principal point and distortion parameters were all similar. For the GoPro camera, the difference in principal points was smaller. However, the differences in the focal length and lens distortion parameters were obviously larger, especially $k_{1}$ and $k_{2}$. These results were very similar to the results for the PH-to-CV transformation. For the parameter of affinity, $b_{1}$ is very small. Based on Equation (7), its influence is quite insignificant compared with other distortion parameters; therefore, it is often ignored in the field of PH.

By using Equations (4)–(6), the correction amounts of lens distortion can be calculated. We checked the correction amounts at four selected points on the original image, and compared the cases using the original ICPs (PH) and the converted ICPs (CV to PH). Table 11 and Table 12 display the numerical results of corrections for Sony and GoPro cameras, respectively. The correction vectors are also shown in Figure 15 and Figure 16, respectively, in which blue dots indicate the selected points and red vectors indicate the corresponding corrections. For both applied cameras, the numerical differences between the compared cases are not significant. This comparison indicates that the proposed method for converting ICPs from CV standard to PH standard is applicable.

## 5. Discussion

Nowadays, image users for measurement, mapping or navigation frequently apply mixed software packages developed in the fields of PH and CV. For example, someone may apply a camera calibration software developed in the field of PH, then apply a CV software to recover image position and orientation. Under this circumstance, parameter transformation is needed to bridge between the software; this paper aims to clarify the camera model definitions and to propose a complete ICP transformation. By taking this approach, image users will be able to apply mixed software; an imagined example of this workflow is depicted in Figure 17. Consequently, further studies and applications could be generated in both fields accordingly. In addition, rectified images can be generated efficiently, no matter if the camera calibration is performed using a PH or CV model.

## 6. Conclusions

This paper expounds the mathematical camera models commonly applied in the fields of PH and CV. The discussion focuses on the different definitions of ICPs, including the focal length, principal point, lens distortion, affinity and shear. This discussion would help researchers to interpret ICPs obtained from the use of software packages developed in both fields.

Based on the discussion, we developed a least-squares adjustment method to transform ICPs between the conventionally used parameters in both fields. This method converts all ICPs usually applied to modern digital cameras, for example, radial distortion, decentring distortion, affinity, and shear parameters. Because both of the transformation models are linear, the calculation process is relatively efficient. The proposed method was verified with a Sony single-lens camera and a GoPro camera, both calibrated with camera calibration software typically used in the fields of PH and CV. Successful transformations of ICPs used in both fields were demonstrated. The accuracy of the transformation may have been slightly affected by the selection of image points. In general, selected image points at regular intervals across the entire image achieved the most favorable results, because this selection strategy accounted for both the number and distribution of selected observation points.

In addition, the comparison with the state of art demonstrates that the proposed method has superior performance. For converting the major parameter of radial distortion (${\hat{k}}_{1}$), the improvement percentages are 91.2% and 91.3% for the Sony and GoPro cameras, respectively. We consequently confirm that our method can improve the conversion of ICPs between models.

Camera calibration software applied in the field of CV commonly performs self-calibration bundle adjustment of overlapped images captured on a specially designed checkerboard. Consequently, the detected feature points are distributed relatively on a plane in the object space. However, coded targets distributed in a three-dimensional space are typically applied in the field of PH. The distribution of the detected object points can be considered the strength of the control space, which will affect the accuracy of derived ICPs. This effect is evident in the test which transformed ICPs obtained using CV software to the ICPs in the field of PH. The rectified images are obviously distorted in the border area of the rectified images. Therefore, where accurate ICPs are concerned, the photogrammetric camera calibration method is preferable to those applications.

## Figures and Tables

**Figure 1 sensors-22-09602-f001:**
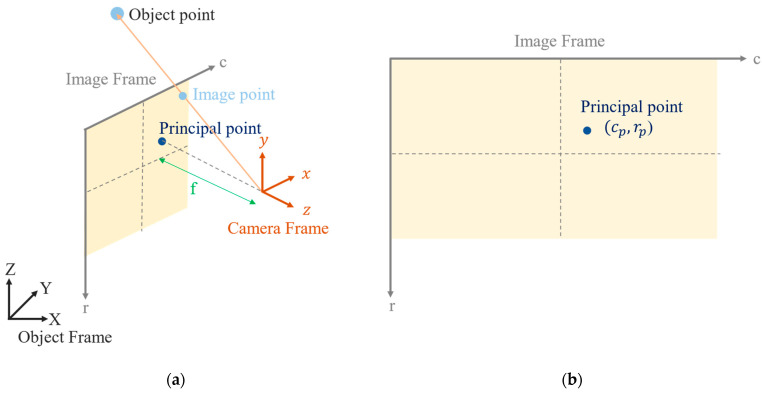
Geometry of perspective projection as defined in the field of PH: (**a**) the three coordinate frames involved and (**b**) the image coordinates of the principal point.

**Figure 2 sensors-22-09602-f002:**
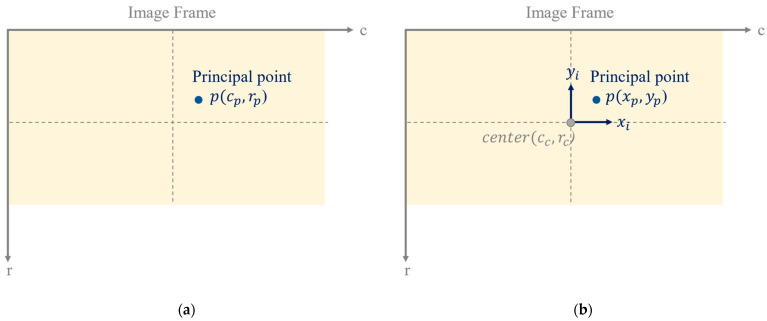
Two definitions of image frame in the field of PH: (**a**) the first: pixel unit, (**b**) the second: metric unit.

**Figure 3 sensors-22-09602-f003:**
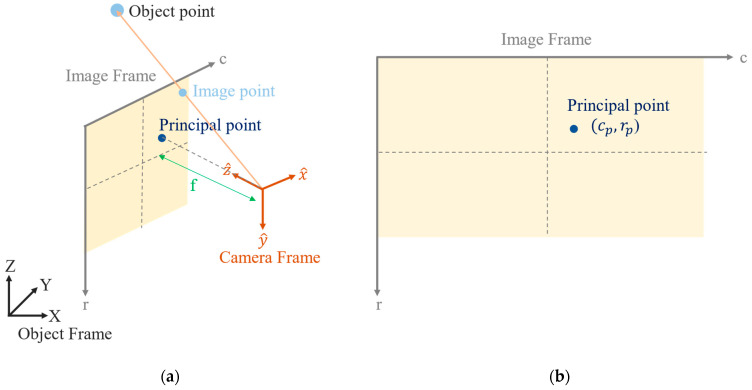
Geometry of perspective projection as defined in the field of CV: (**a**) the three coordinate frames involved and (**b**) the image coordinates of the principal point.

**Figure 4 sensors-22-09602-f004:**
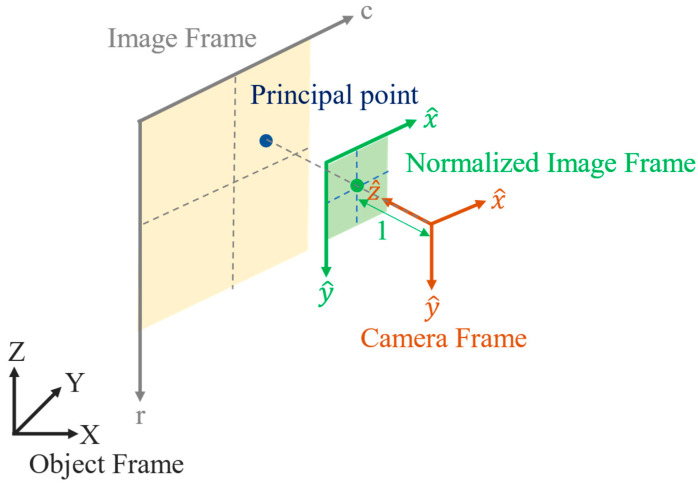
Geometry of normalized image frame in the field of CV.

**Figure 5 sensors-22-09602-f005:**
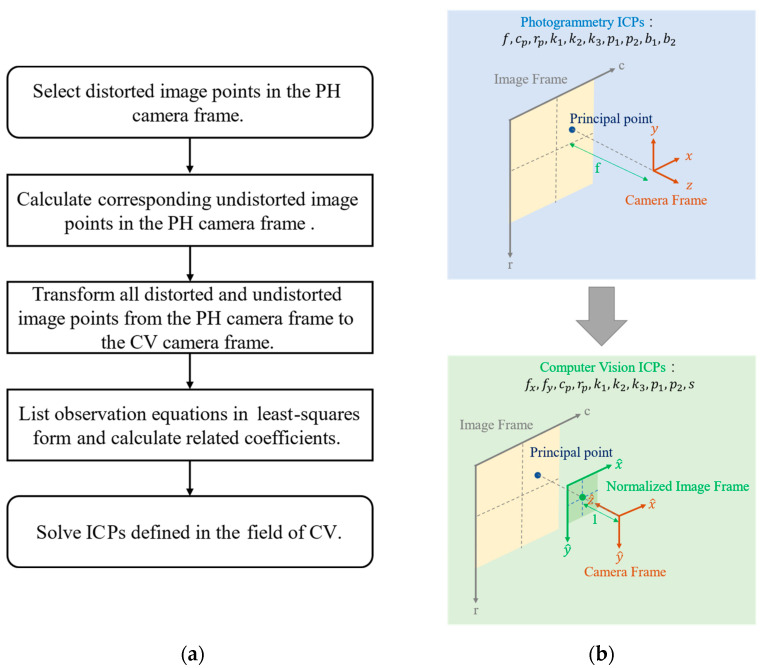
Workflow of transformation of PH ICPs to CV ICPs: (**a**) the description of each step and (**b**) visualized framework.

**Figure 6 sensors-22-09602-f006:**
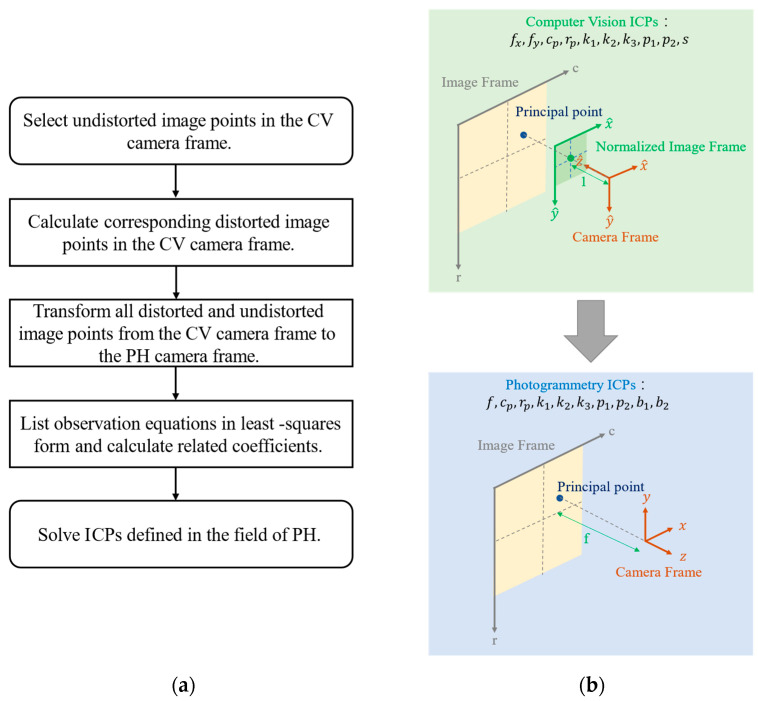
Workflow of transformation of CV ICPs to PH ICPs: (**a**) the description of each step and (**b**) visualized framework.

**Figure 7 sensors-22-09602-f007:**
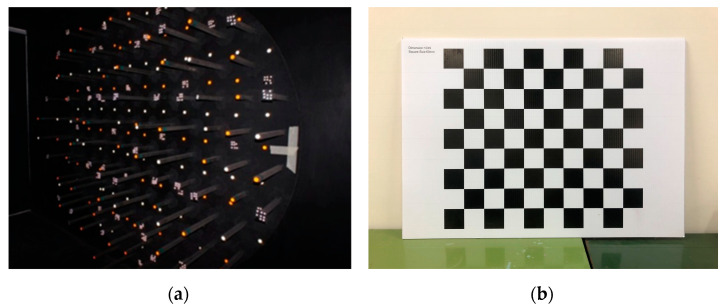
Camera calibration method tools: (**a**) rotated table with coded targets for the PH method; (**b**) checkerboard for the CV method.

**Figure 8 sensors-22-09602-f008:**
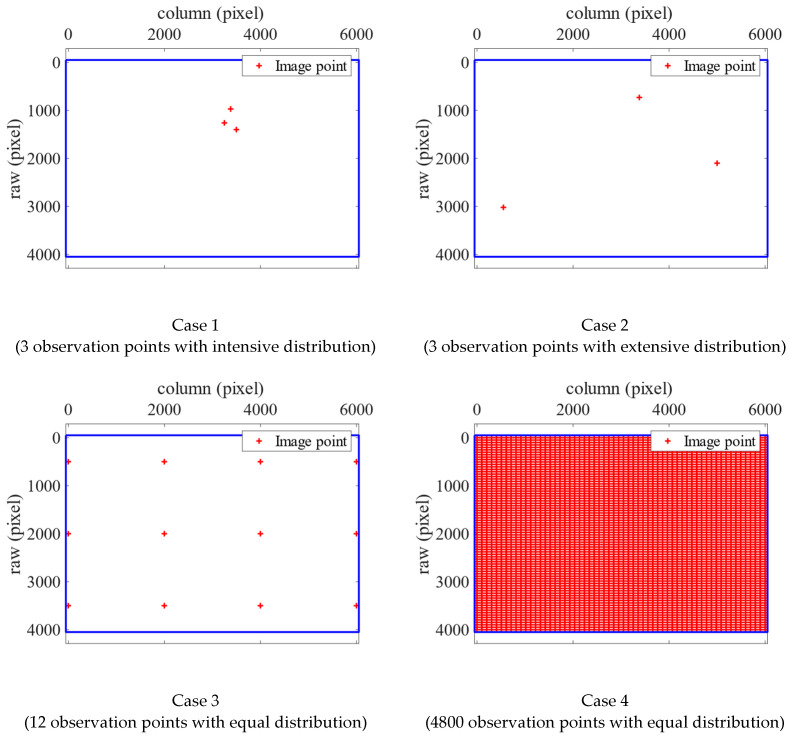
Selected observation points in Case 1 to 4 (Sony).

**Figure 9 sensors-22-09602-f009:**
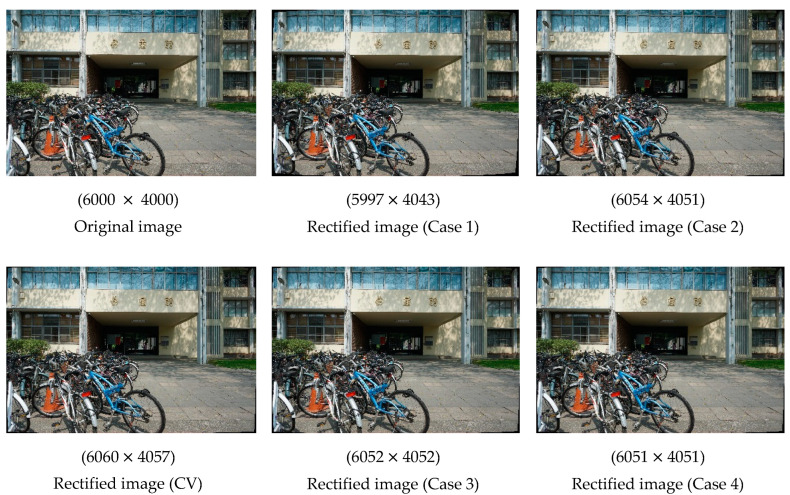
Rectified images generated using different methods (Sony).

**Figure 10 sensors-22-09602-f010:**
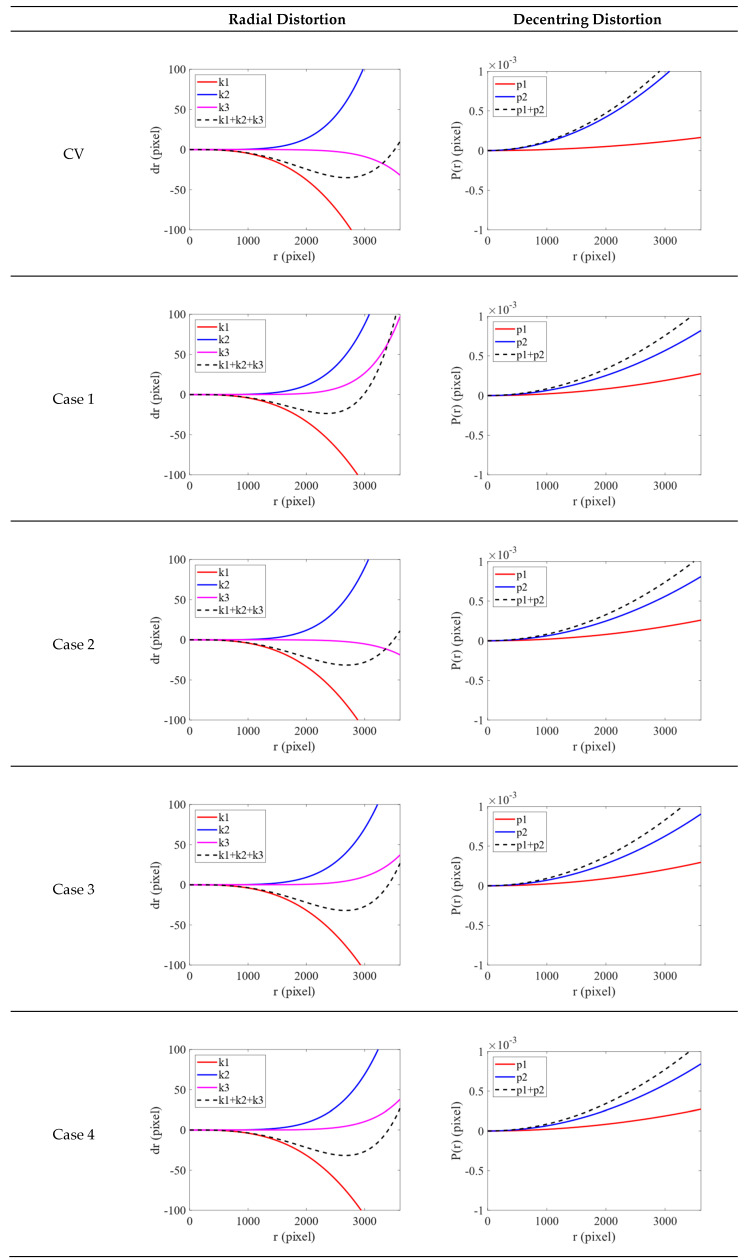
Radial distortion and decentring distortion plots (Sony).

**Figure 11 sensors-22-09602-f011:**
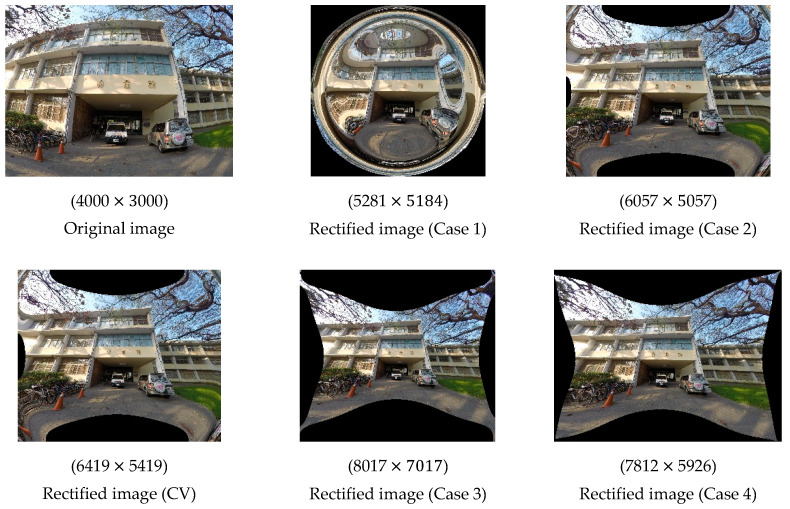
Rectified images generated using different methods (GoPro).

**Figure 12 sensors-22-09602-f012:**
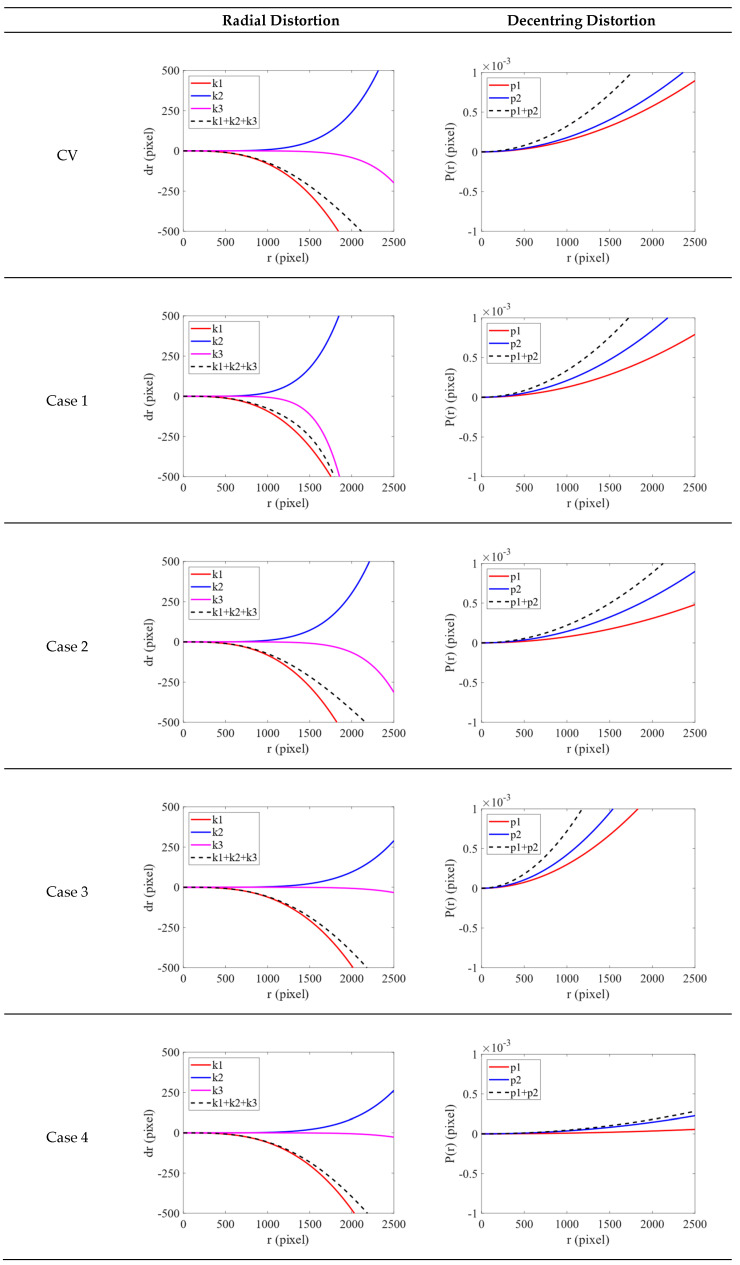
Radial distortion and decentring distortion plots (GoPro).

**Figure 13 sensors-22-09602-f013:**
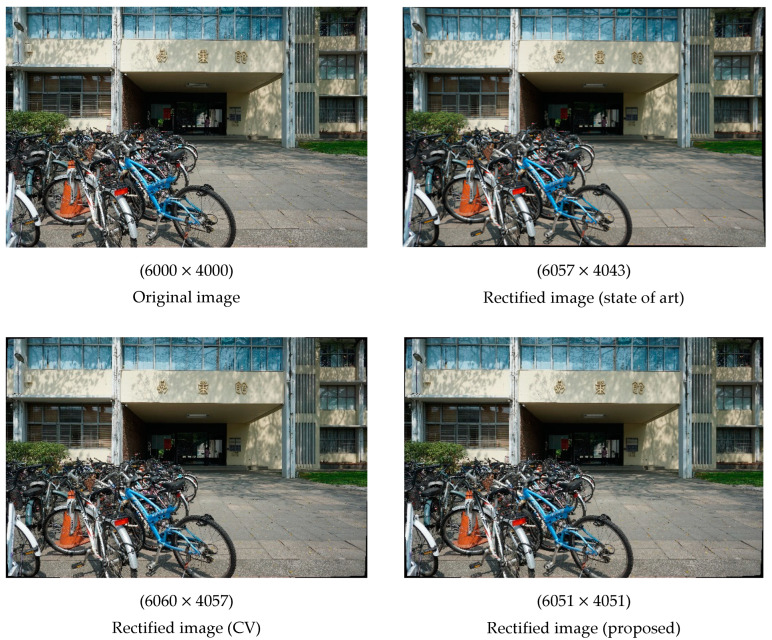
Comparison of rectified images (Sony).

**Figure 14 sensors-22-09602-f014:**
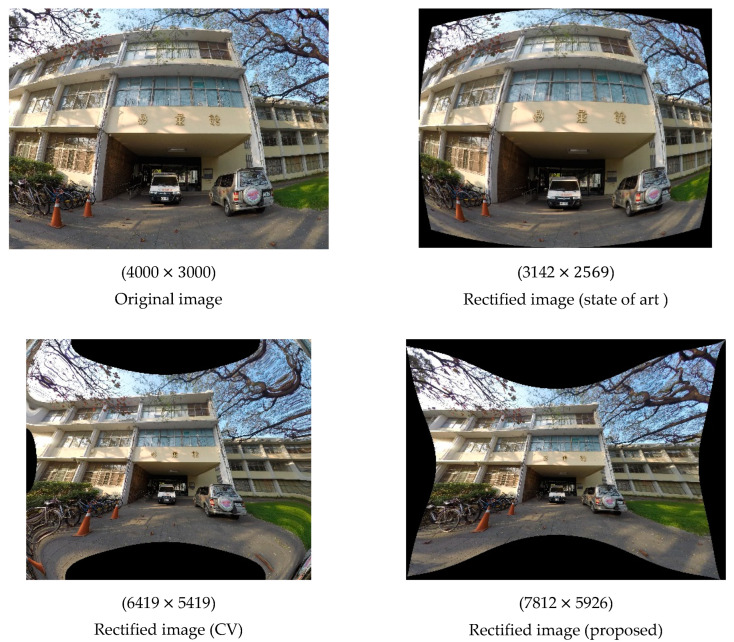
Comparison of rectified images (GoPro).

**Figure 15 sensors-22-09602-f015:**
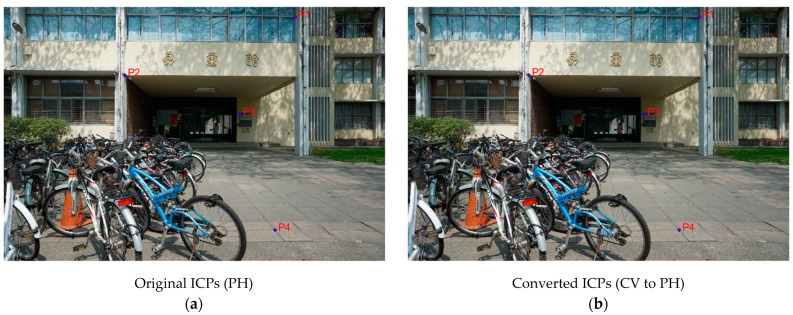
Visual results of corrections (Sony): (**a**) using the original ICPs (PH) (**b**) using the converted ICPs (CV to PH).

**Figure 16 sensors-22-09602-f016:**
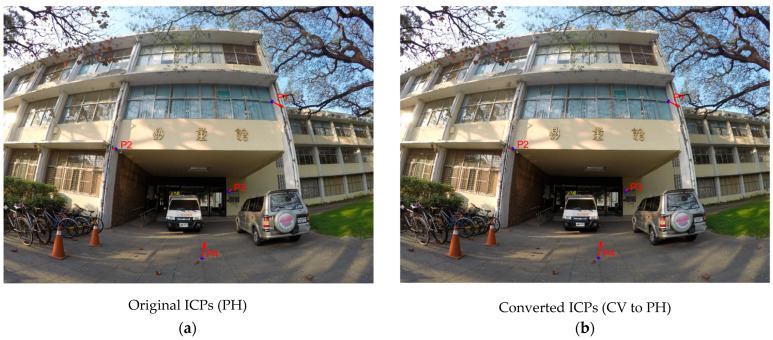
Visual results of corrections (GoPro): (**a**) using PH ICPs (**b**) using CV to PH ICPs.

**Figure 17 sensors-22-09602-f017:**
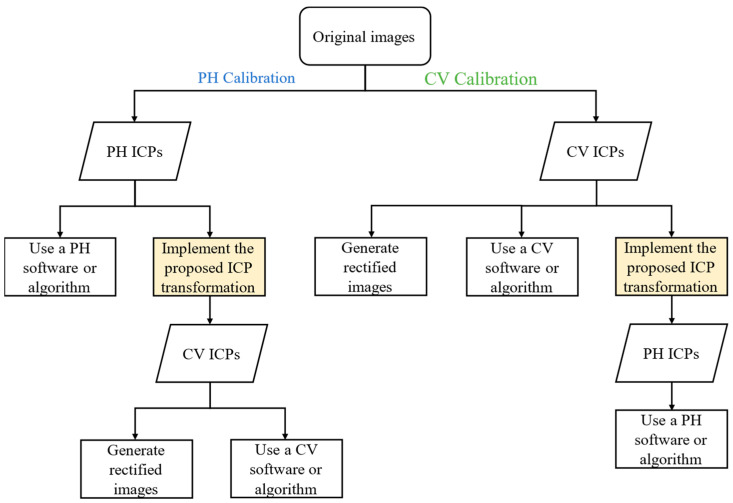
Imagined workflow for using mixed software applied in the fields of PH and CV.

**Table 1 sensors-22-09602-t001:** Symbols and corresponding meaning in the mathematical expressions.

Photogrammetry		Computer Vision	
Symbol	Meaning	Symbol	Meaning
f	Focal length.	$f_{x}$	Focal length related to x direction.
		$f_{y}$	Focal length related to y direction.
(c, r)	Coordinates of image point in the image frame: pixel unit.	(c, r)	Coordinates of image point in the image frame: pixel unit.
($c_{p}$*,* $r_{p}$)	Coordinates of principal point in the image frame: pixel unit.	($c_{p}$*,* $r_{p}$)	Coordinates of principal point in the image frame: pixel unit.
($c_{c}$*,* $r_{c}$)	Coordinates of image center in the image frame: pixel unit.	($c_{c}$*,* $r_{c}$)	Coordinates of image center in the image frame: pixel unit.
($x_{i}$*,* $y_{i}$)	Coordinates of image point in another image frame: metric unit.		
($x_{p}$*,* $y_{p}$)	Coordinates of principal point in another image frame: metric unit.		
$k_{1}$, $k_{2}$, $k_{3}$	Parameters of radial distortion.	${\hat{k}}_{1}$, ${\hat{k}}_{2}$, ${\hat{k}}_{3}$	Parameters of radial distortion.
$p_{1}$, $p_{2}$	Parameters of decentring distortion	${\hat{p}}_{1}$, ${\hat{p}}_{2}$	Parameters of decentring distortion.
$b_{1}$	Parameter of affinity.		
$b_{2}$	Parameter of shear	s	Parameter of shear (skew)
ds	Image pixel size.	$ds_{x}$	Image pixel size in the x direction.
		$ds_{y}$	Image pixel size in the y direction.
(x, y)	Coordinates of image point in the camera frame without considering lens distortion.	$(\hat{x},\;\hat{y}$)	Coordinates of image point in the camera frame without considering lens distortion.
$(x_{d},\;y_{d}$)	Coordinates of distorted image points in the camera frame.	(${\hat{x}}_{d},\;\;{\hat{y}}_{d}$)	Coordinates of distorted image points in the camera frame.
($x_{u},\;y_{u}$)	Coordinates of undistorted image points in the camera frame.	(${\hat{x}}_{u},\;\;{\hat{y}}_{u}$)	Coordinates of undistorted image points in the camera frame.
$\Delta x_{rad}\left(x_{d},y_{d}\right)$, $\Delta y_{rad}\left(x_{d},y_{d}\right)$	Radial distortion model formed by distorted image points.	$\Delta {\hat{x}}_{rad}\left({\hat{x}}_{u},{\hat{y}}_{u}\right)$, $\Delta {\hat{y}}_{rad}\left({\hat{x}}_{u},{\hat{y}}_{u}\right)$	Radial distortion model formed by undistorted image points.
$\Delta x_{dec}\left(x_{d},y_{d}\right)$, $\Delta y_{dec}\left(x_{d},y_{d}\right)$	Decentring distortion model formed by distorted image points.	$\Delta {\hat{x}}_{dec}\left({\hat{x}}_{u},{\hat{y}}_{u}\right)$, $\Delta {\hat{y}}_{dec}\left({\hat{x}}_{u},{\hat{y}}_{u}\right)$	Decentring distortion model formed by undistorted image points.
$\Delta x_{aff}\left(x_{d},y_{d}\right)$, $\Delta y_{aff}\left(x_{d},y_{d}\right)$	Affine distortion model formed by distorted image points.		
r	In the camera frame, the distance from distorted image points to principal point.	$\hat{r}$	In the camera frame, the distance from undistorted image points to principal point.

**Table 2 sensors-22-09602-t002:** Camera specifications.

Specification	Sony A6000	GoPro Hero 4
Focal length (mm)	16	2.8
Field of view (degree)	72	94
Image resolution (pixel)	6000 × 4000	4000 × 3000
Pixel size (mm)	0.0039	0.0015

**Table 3 sensors-22-09602-t003:** Sony A6000 camera calibration.

Photogrammetry			Computer Vision		
ICPs	Value	Std.	ICPs	Value	Std.
$f\left(mm\right)$	15.8657	±0.002	$f_{x}\left(pixel\right)$	4076.82	±1.43
$c_{0}\left(pixel\right)$	2962.49	±0.26	$f_{y}\left(pixel\right)$	4079.62	±1.43
$r_{0}\left(pixel\right)$	1961.21	±0.26	$c_{0}\left(pixel\right)$	2957.94	±1.41
$k_{1}\left(mm^{-2}\right)$	$2.77\times 10^{-4}$	± $8.63\times 10^{-7}$	$r_{0}\left(pixel\right)$	1966.85	±1.46
$k_{2}\left(mm^{-4}\right)$	$-1.51\times 10^{-6}$	± $1.18\times 10^{-8}$	${\hat{k}}_{1}\left(unit\right)$	−0.0782	± $8.51\times 10^{-4}$
$k_{3}\left(mm^{-6}\right)$	$-3.15\times 10^{-10}$	± $4.82\times 10^{-11}$	${\hat{k}}_{2}\left(unit\right)$	0.1190	± $3.46\times 10^{-3}$
$p_{1}\left(mm^{-1}\right)$	$-2.62\times 10^{-5}$	± $9.22\times 10^{-7}$	${\hat{k}}_{3}\left(unit\right)$	−0.0185	$\pm 3.75\times 10^{-3}$
$p_{2}\left(mm^{-1}\right)$	$1.17\times 10^{-5}$	± $6.86\times 10^{-7}$	${\hat{p}}_{1}\left(unit\right)$	$-1.13\times 10^{-4}$	± $1.02\times 10^{-4}$
$b_{1}$	0	-	${\hat{p}}_{2}\left(unit\right)$	$6.87\times 10^{-4}$	± $9.93\times 10^{-5}$
$b_{2}$	0	-	$s\left(unit\right)$	0	-

**Table 4 sensors-22-09602-t004:** GoPro Hero 4 camera calibration.

Photogrammetry			Computer Vision		
ICPs	Value	Std.	ICPs	Value	Std.
$f\left(mm\right)$	2.7321	±0.001	$f_{x}\left(pixel\right)$	1753.97	±1.37
$c_{0}\left(pixel\right)$	1930.20	±0.67	$f_{y}\left(pixel\right)$	1757.67	±1.33
$r_{0}\left(pixel\right)$	1534.07	±0.67	$c_{0}\left(pixel\right)$	1925.04	±0.92
$k_{1}\left(mm^{-2}\right)$	0.0412	± $1.71\times 10^{-5}$	$r_{0}\left(pixel\right)$	1533.72	±0.96
$k_{2}\left(mm^{-4}\right)$	$3.95\times 10^{-4}$	± $3.69\times 10^{-6}$	${\hat{k}}_{1}\left(unit\right)$	−0.2460	$\pm 4.99\times 10^{-4}$
$k_{3}\left(mm^{-6}\right)$	$-1.84\times 10^{-4}$	± $2.48\times 10^{-7}$	${\hat{k}}_{2}\left(unit\right)$	0.0711	$\pm 4.02\times 10^{-4}$
$p_{1}\left(mm^{-1}\right)$	$1.16\times 10^{-4}$	± $2.16\times 10^{-6}$	${\hat{k}}_{3}\left(unit\right)$	−0.0095	± $9.51\times 10^{-5}$
$p_{2}\left(mm^{-1}\right)$	$8.66\times 10^{-5}$	± $1.88\times 10^{-6}$	${\hat{p}}_{1}\left(unit\right)$	$-2.24\times 10^{-4}$	± $5.27\times 10^{-5}$
$b_{1}$	0	-	${\hat{p}}_{2}\left(unit\right)$	$-2.23\times 10^{-4}$	± $4.82\times 10^{-5}$
$b_{2}$	0	-	$s\left(unit\right)$	0	-

**Table 5 sensors-22-09602-t005:** Case 1 to 4 transformation results (Sony).

ICPs	CV	Case 1	Case 2	Case 3	Case 4
$f_{x}\left(pixel\right)$	4076.82	4068.13	4068.13	4068.13	4068.13
$f_{y}\left(pixel\right)$	4079.62	4068.13	4068.13	4068.13	4068.13
$c_{0}\left(pixel\right)$	2957.94	2962.49	2962.49	2962.49	2962.49
$r_{0}\left(pixel\right)$	1966.85	1961.21	1961.21	1961.21	1961.21
${\hat{k}}_{1}\left(unit\right)$	−0.0782	−0.0692	−0.0691	−0.0658	−0.0652
${\hat{k}}_{2}\left(unit\right)$	0.1190	0.0989	0.1013	0.0790	0.0779
${\hat{k}}_{3}\left(unit\right)$	−0.0185	0.0555	−0.0107	0.0213	0.0217
${\hat{p}}_{1}\left(unit\right)$	$-1.13\times 10^{-4}$	$1.88\times 10^{-4}$	$1.78\times 10^{-4}$	$2.02\times 10^{-4}$	$1.87\times 10^{-4}$
${\hat{p}}_{2}\left(unit\right)$	$6.87\times 10^{-4}$	$4.08\times 10^{-4}$	$4.03\times 10^{-4}$	$4.50\times 10^{-4}$	$4.20\times 10^{-4}$
$s\left(unit\right)$	0	0	0	0	0

**Table 6 sensors-22-09602-t006:** Case 1 to 4 transformation results (GoPro).

ICPs	CV	Case 1	Case 2	Case 3	Case 4
$f_{x}\left(pixel\right)$	1753.97	1821.40	1821.40	1821.40	1821.40
$f_{y}\left(pixel\right)$	1757.67	1821.40	1821.40	1821.40	1821.40
$c_{0}\left(pixel\right)$	1925.04	1930.20	1930.20	1930.20	1930.20
$r_{0}\left(pixel\right)$	1533.72	1534.07	1534.07	1534.07	1534.07
${\hat{k}}_{1}\left(unit\right)$	−0.2460	−0.3075	−0.2736	−0.2027	−0.1981
${\hat{k}}_{2}\left(unit\right)$	0.0711	0.2555	0.1042	0.0326	0.0296
${\hat{k}}_{3}\left(unit\right)$	−0.0095	−0.2415	−0.0188	−0.0020	−0.0016
${\hat{p}}_{1}\left(unit\right)$	$2.24\times 10^{-4}$	$2.13\times 10^{-4}$	$1.30\times 10^{-4}$	$-5.02\times 10^{-4}$	$-1.49\times 10^{-5}$
${\hat{p}}_{2}\left(unit\right)$	$2.23\times 10^{-4}$	$-2.82\times 10^{-4}$	$-1.93\times 10^{-4}$	$5.65\times 10^{-4}$	$4.75\times 10^{-5}$
$s\left(unit\right)$	0	0	0	0	0

**Table 7 sensors-22-09602-t007:** Comparison of transformation results (Sony).

ICPs	CV	PH to CV (State of Art)	PH to CV (Proposed)
$f_{x}\left(pixel\right)$	4076.82	4068.13	4068.13
$f_{y}\left(pixel\right)$	4079.62	4068.13	4068.13
$c_{0}\left(pixel\right)$	2957.94	2962.49	2962.49
$r_{0}\left(pixel\right)$	1966.85	1961.21	1961.21
${\hat{k}}_{1}\left(unit\right)$	−0.0782	0.0696	−0.0652
${\hat{k}}_{2}\left(unit\right)$	0.1190	−0.0959	0.0779
${\hat{k}}_{3}\left(unit\right)$	−0.0185	−0.0050	0.0217
${\hat{p}}_{1}\left(unit\right)$	$-1.13\times 10^{-4}$	$-1.85\times 10^{-4}$	$1.87\times 10^{-4}$
${\hat{p}}_{2}\left(unit\right)$	$6.87\times 10^{-4}$	$-4.15\times 10^{-4}$	$4.20\times 10^{-4}$

**Table 8 sensors-22-09602-t008:** Comparison of transformation results (GoPro).

ICPs	CV	PH to CV (State of Art)	PH to CV (Proposed)
$f_{x}\left(pixel\right)$	1753.97	1821.40	1821.40
$f_{y}\left(pixel\right)$	1757.67	1821.40	1821.40
$c_{0}\left(pixel\right)$	1925.04	1930.20	1930.20
$r_{0}\left(pixel\right)$	1533.72	1534.07	1534.07
${\hat{k}}_{1}\left(unit\right)$	−0.2460	0.3074	−0.1981
${\hat{k}}_{2}\left(unit\right)$	0.0711	0.0220	0.0296
${\hat{k}}_{3}\left(unit\right)$	−0.0095	0.0766	−0.0016
${\hat{p}}_{1}\left(unit\right)$	$2.24\times 10^{-4}$	$-2.37\times 10^{-4}$	$-1.49\times 10^{-5}$
${\hat{p}}_{2}\left(unit\right)$	$2.23\times 10^{-4}$	$3.17\times 10^{-4}$	$4.75\times 10^{-5}$

**Table 9 sensors-22-09602-t009:** Overall transformation results (Sony).

ICPs	PH	CV to PH	Difference
$f\left(mm\right)$	15.8657	15.9105	0.0448
$c_{0}\left(pixel\right)$	2962.49	2957.94	−4.55
$r_{0}\left(pixel\right)$	1961.21	1966.85	5.64
$k_{1}\left(mm^{-2}\right)$	$2.77\times 10^{-4}$	$3.25\times 10^{-4}$	$4.89\times 10^{-5}$
$k_{2}\left(mm^{-4}\right)$	$-1.51\times 10^{-6}$	$-2.09\times 10^{-6}$	$-5.81\times 10^{-7}$
$k_{3}\left(mm^{-6}\right)$	$-3.15\times 10^{-10}$	$1.91\times 10^{-9}$	$2.22\times 10^{-9}$
$p_{1}\left(mm^{-1}\right)$	$-2.62\times 10^{-5}$	$-4.35\times 10^{-5}$	$-1.74\times 10^{-5}$
$p_{2}\left(mm^{-1}\right)$	$1.17\times 10^{-5}$	$-7.31\times 10^{-6}$	$-1.90\times 10^{-5}$
$b_{1}$	0	$-7.25\times 10^{-6}$	$-7.25\times 10^{-6}$
$b_{2}$	0	0	0

**Table 10 sensors-22-09602-t010:** Overall transformation results (GoPro).

ICPs	PH	CV to PH	Difference
$f\left(mm\right)$	2.7321	2.6365	−0.0956
$c_{0}\left(pixel\right)$	1930.20	1925.04	−5.16
$r_{0}\left(pixel\right)$	1534.07	1533.72	−0.35
$k_{1}\left(mm^{-2}\right)$	0.0412	0.0323	−0.0089
$k_{2}\left(mm^{-4}\right)$	$3.95\times 10^{-4}$	0.0042	0.0038
$k_{3}\left(mm^{-6}\right)$	$1.84\times 10^{-4}$	$-1.31\times 10^{-4}$	$-3.16\times 10^{-4}$
$p_{1}\left(mm^{-1}\right)$	$1.16\times 10^{-4}$	$-2.34\times 10^{-4}$	$-3.51\times 10^{-4}$
$p_{2}\left(mm^{-1}\right)$	$8.66\times 10^{-5}$	$-2.23\times 10^{-4}$	$-3.09\times 10^{-4}$
$b_{1}$	0	$4.03\times 10^{-4}$	$4.03\times 10^{-4}$
$b_{2}$	0	0	0

**Table 11 sensors-22-09602-t011:** Numerical results of corrections (Sony).

	Original ICPs (PH)			Converted ICPs (CV to PH)		
Image Point	Correction in x Direction (mm)	Correction in y Direction (mm)	Total Corrections on the Image (Pixel)	Correction in x Direction (mm)	Correction in y Direction (mm)	Total Corrections on the Image (Pixel)
P1	0.0751	0.0868	29.44	0.0790	0.0911	30.92
P2	−0.0298	0.0247	9.93	−0.0343	0.0277	11.31
P3	0.0068	0.0026	1.88	0.0077	0.0028	2.09
P4	0.0539	−0.0653	21.71	0.0600	−0.0745	24.53

**Table 12 sensors-22-09602-t012:** Numerical results of corrections (GoPro).

	Original ICPs (PH)			Converted ICPs (CV to PH)		
Image Point	Correction in x Direction (mm)	Correction in y Direction (mm)	Total Corrections on the Image (Pixel)	Correction in x Direction (mm)	Correction in y Direction (mm)	Total Corrections on the Image (Pixel)
P1	0.1733	0.0891	129.91	0.1697	0.0864	126.93
P2	−0.0513	−0.0008	34.21	−0.0455	−0.0010	30.36
P3	0.0350	−0.0324	31.79	0.0310	−0.0292	28.40
P4	0.0480	−0.2633	178.42	0.0489	−0.2691	182.33

## Data Availability

Not applicable.

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
