# Peer review of "Interpretation and Transformation of Intrinsic Camera Parameters Used in Photogrammetry and Computer Vision"

_sensors, 2022, doi:10.3390/s22249602_

Round 1
Reviewer 1 Report
In the manuscript, The least square adjustment method for ICP transformation in PH and CV image frames has been proposed. The methodology has been explained well and mathematical expressions are sufficient to understand the formulated problem. The benefits of the proposed method have been explained without a doubt. However, I realized a technical issue regarding the comparison between the proposed methodology and the state of the art. There is no numerical or performance comparison between the results in the manuscript and the results in the literature. There isn't even any baseline performance to demonstrate the clear efficiency of the methodology in the manuscript.
All my suggestions regarding the manuscript are listed below:
- All figures should be bigger to follow all details about the methodology and results.
- The state-of-the-art performance should be given or baseline performance should be added at least.
- Parameters, variables, and constants in all mathematical expressions must be clearly explained in the text. I strongly suggest adding a table explaining all unclear parameters, variables, and constants.
- Figures 5 and 6 should be extended by adding meaningful visual materials to clarify all processes more.
-Conclusion should contain numerical comparisons between the proposed methodology and state-of-the-art to demonstrate performance improvement.
Author Response
Dear Reviewer,
Thank you very much for the comments concerning our manuscript. Those comments are all valuable and very helpful for revising and improving this manuscript paper. We have studied comments carefully and have adopted correction according to those suggestions. The corrections in the manuscript are marked up using the “Track changes” function, and the responds to the comments are written in the report.
Please see the attachment.

Reviewer 2 Report
This paper provides a clear presentation of a least-squares optimization approach to transforming intrinsic camera parameters from one standard mathematical representation to another, which allows interoperability between different software packages that may utilize different standards. The approach would conceivably be useful in some applications where performing new camera calibrations may not be feasible, or where a photogrammetry-based calibration may provide better distortion correction accuracy but needs to be converted to a computer vision representation.
The results are clearly presented, and the experiments seem to justify the method well. My only concern with the paper is that the conclusions stated in 4.3 -- "the converted ICPs were approximately equal to the original ICPs" -- is rather subjective. Many of the parameter values differ by a large percentage (for example, in Table 7, parameter k2 goes from 0.000395 to 0.0042, which is more than an order of magnitude different). While the absolute values, and thus the absolute differences, are small compared to the other parameters, small changes in these higher-order coefficients can have a much more pronounced effect on the image transformation than in other parameters. How do we know that the values are "approximately equal"? I don't think this judgment can be made without visually inspecting the results of these transformation parameters on a sample image. This conclusion would be justified by adding image results to the paper, as was done in Section 4.2.
Author Response

(The authors gave the same response as above.)

Round 2
Reviewer 1 Report
Table 1 should be placed in 2. Camera Mathematical Model so all variables and parameters will be clear enough for all. If numerical improvement based on the state of the art is added, it will be helpful to evaluate how much some parameters improve and how the final performance has been affected by this improvement.
Author Response
Thank you for the comments. Table 1 has been placed in the section of Camera Mathematical Model (page 9).
The numerical improvement when using the proposed method has been added to the revised manuscript (page 23). The abstract and conclusion have been also revised accordingly.
We sincerely appreciate your insightful comments and constructive suggestions. In addition, we feel grateful that you spent so much time and patience in reviewing our manuscript entitled “Interpretation and Transformation of Intrinsic Camera Parameters Used in Photogrammetry and Computer Vision”.